# Generic residue numbering of the GAIN domain of adhesion GPCRs

Florian Seufert [1], Guillermo Pérez-Hernández [2], Gáspár Pándy-Szekeres [3,4], Ramon Guixà-González [2,9], Tobias Langenhan [5,6,7], David E. Gloriam [3] ✉ & Peter W. Hildebrand [1,8] ✉

The GPCR autoproteolysis inducing (GAIN) domain is an ancient protein fold ubiquitous in adhesion G protein-coupled receptors (aGPCR). It contains a tethered agonist necessary and sufficient for receptor activation. The GAIN domain is a hotspot for pathological mutations. However, the low primary sequence conservation of GAIN domains has thus far hindered the knowledge transfer across different GAIN domains in human receptors as well as species orthologs. Here, we present a scheme for generic residue numbering of GAIN domains, based on structural alignments of over 14,000 modeled GAIN domain structures. This scheme is implemented in the GPCR database (GPCRdb) and elucidates the domain topology across different aGPCRs and their homologs in a large panel of species. We identify conservation hotspots and statistically cancer-enriched positions in human aGPCRs and show the transferability of positional and structural information between GAIN domain homologs. The GAIN-GRN scheme provides a robust strategy to allocate structural homologies at the primary and secondary levels also to GAIN domains of polycystic kidney disease 1/PKD1-like proteins, which now renders positions in both GAIN domain types comparable to one another. In summary, our work enables researchers to generate hypothesis and rationalize experiments related to GAIN domain function and pathology.

Adhesion/class B2 G protein-coupled receptors (aGPCRs), the second-largest class of GPCRs, have garnered substantial research and medical interest due to their involvement in neural development, hereditary disorders, and cancers among others[1–4]. aGPCRs are classed into nine subfamilies[5] and are characterized by a very large extracellular region, containing the conserved GPCR autoproteolysis inducing (GAIN) domain. The GAIN domain is positioned directly N-terminal of the seven-transmembrane domain (7TM,

Fig. 1a), which transduces an extracellular signal to intracellular effector proteins[6].

The GAIN domain serves several functions. First, at its GPCR proteolysis site (GPS) an autoproteolytic cleavage event occurs adjacent to the 7TM domain, which yields a bipartite structure stabilized by non-covalent interactions[7–10]. The two resulting elements, called N-terminal and C-terminal fragments (NTF/CTF), remain attached to one another even at the cell surface. Second, the GAIN domain

[1]Institute for Medical Physics and Biophysics, Leipzig University, Medical Faculty, Leipzig, Germany. [2]Institute for Medical Physics and Biophysics, Charité – Universitätsmedizin Berlin, corporate member of Freie Universität Berlin and Humboldt-Universität zu Berlin, Berlin, Germany. [3]Department of Drug Design and Pharmacology, University of Copenhagen, Universitetsparken 2, Copenhagen, Denmark. [4]Medicinal Chemistry Research Group, HUN-REN Research Center for Natural Sciences, Magyar Tudósok körútja 2., Budapest, Hungary. [5]Rudolf Schönheimer Institute of Biochemistry, Division of General Biochemistry, Medical Faculty, Leipzig University, Leipzig, Germany. [6]Comprehensive Cancer Center Central Germany (CCCG), Leipzig, Germany. [7]Institute of Biology, Faculty of Life Sciences, Leipzig University, Leipzig, Germany. [8]Center for Scalable Data Analytics and Artificial Intelligence (ScaDS.AI), Leipzig, Germany. [9]Present address: Department of Biological Chemistry, Institute for Advanced Chemistry of Catalonia (IQAC-CSIC), Barcelona, Spain. ✉e-mail: david.gloriam@sund.ku.dk; peter.hildebrand@medizin.uni-leipzig.de

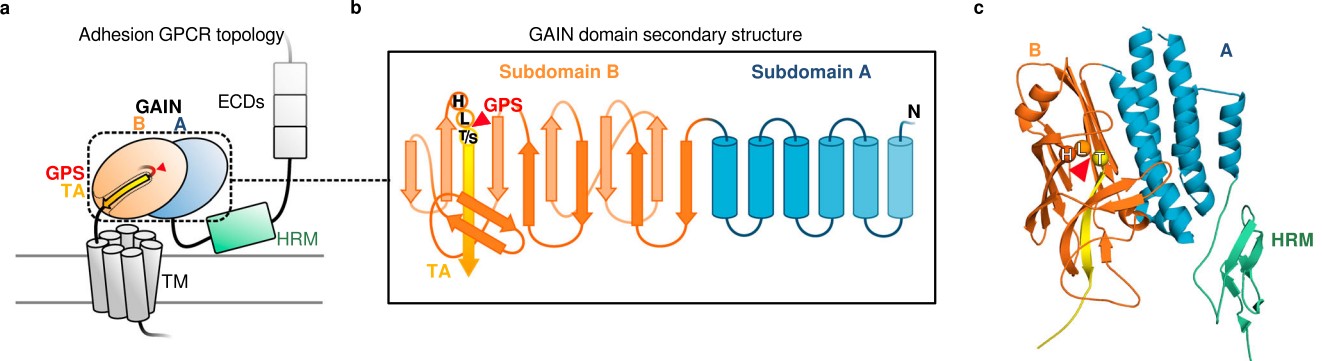

**Fig. 1 | The variable topology of the G protein-coupled receptor (GPCR) autoproteolysis inducing (GAIN) domain. a** adhesion GPCR topology with the N-terminal extracellular region composed of various extracellular domains (ECDs) and the GAIN domain directly N-terminal of the seven-transmembrane domain (TM). The GAIN domain is comprised of subdomains A and B, with the tethered agonist (TA) as the most C-terminal β-strand. The GAIN domain is frequently preceded by a hormone receptor motif (HRM) domain of unknown function. **b** the GAIN domain is composed of two subdomains, with subdomain A (blue) comprised of 2–6 α-helices with conservation decreasing toward the N-terminal boundary. The β-sandwich subdomain B (orange) is composed of 13–14 β-strands with a conserved autoproteolytic cleavage triad (GPS) of sequence HL | T/S (red triangle: cleavage site), followed by the tethered agonist (TA, yellow). **c** the GAIN domain of rat ADGRL1 (PDB ID: 4DLQ[9]) shows all hallmarks of the GAIN domain.

contains a tethered agonist element (TA, Stachel, Fig. 1b), which corresponds to the N-terminus of the CTF that arises through GAIN domain cleavage[11–14]. The TA activates the receptor upon dissociation of the NTF/CTF complex[11,12,15–17], the biophysical intricacies of which are yet to be uncovered[18,19]. Third, several aGPCRs act as metabotropic mechanosensors[20–24], where the GAIN domain is proposed to serve as a molecular integrator of mechanical forces through its partial unfolding or eventual dissociation of the NTF/CTF complex upon force stimulation[25–29].

X-ray and cryo-EM structures provided the first insights into GAIN domain structures and TA-7TM complexes[9,10,30–36]. The available set of experimentally determined GAIN domain structures indicates a common architecture with two, structurally variable, subdomains: The more variable subdomain A is comprised of up to six helices, and the more sequence-conserved subdomain B adopts a β-sandwich with the TA as its most C-terminal strand (Fig. 1b, c)[9,10,19,35–37]. The low sequence identity of GAIN domains and variable number of constituting segments, however, led to inadequate annotations of the GAIN domain in protein databases, hampering inter-species comparison of GAIN domains and limiting a holistic understanding of GAIN domain function.

Generic residue numbers (GRNs) provide a common index to corresponding amino acids across the different members of a protein family. GRNs have a great utility as they enable comparison and inference of a multitude of residue data spanning pharmacology (e.g., in vitro mutations), structural biology (e.g., ligand, domain, or protein interactions), and genetics (natural variants). For GPCRs, the first GRN scheme was that of *Ballesteros-Weinstein* and assigned residue indices in the 7TM domains of class A GPCRs[38]. A number, 50, is given to the most conserved residue in each of the seven helices that serve as a reference when assigning consecutive numbers of upstream and downstream. This system has since been adapted to other GPCR classes[39], including the Wootten numbering scheme for the class B1 (Secretin) and B2 (Adhesion) receptor families[40].

As GPCR structures became available, these sequence-based schemes were found to suffer from non-generic numbers when some receptors have helix bulges or constrictions causing a one-position residue gap in structural alignment and offset of following residues in sequence alignment[39,41]. To mitigate this numbering issue, the GPCR database (GPCRdb) provided structure-based GRN schemes for each GPCR class wherein structural residue gaps are also present as single gaps in the sequence alignment[39]. The GPCRdb schemes also added helix 8 (H8) and structurally conserved stretches of the first extra- and intracellular loops. Highly flexible and variable protein regions such as

loops remain unannotated in currently established schemes. GRNs have seen wide adoption among researchers, becoming a frequently used tool in communicating GPCR research. Other GRN schemes like the kinase –ligand interaction fingerprints and structure database (KLIFS)[42–44], the common G protein Gα numbering (CGN)[45], or the common arrestin numbering (CAN)[46] serve in mapping functional protein networks or drug-development purposes in additional protein families.

Here, we introduce a GRN scheme for aGPCR GAIN domains based on the superposition of more than 14,000 structural GAIN domain models generated with ColabFold/AlphaFold 2[47,48]. We highlight structural variability and common features of all aGPCR GAIN domains and enable data transfer by means of common GRN labels by finding statistically cancer-enriched positions in humans. The GAIN-GRN was implemented into the GPCRdb[41,49–51] to allow for intuitive use in a highly accessible and widely adapted resource, as well as for programmatic access to the data. In addition, we provide a notebook in the code repository (https://github.com/FloSeu/GAIN-GRN) enabling the ad-hoc indexing of any GAIN-domain containing protein. We expect that these will inspire future studies aiming to elucidate the molecular mechanism of GAIN domains in the signal transduction and physiological functions of aGPCRs, and will aid analyses on how structural anomalies contribute to aGPCR dysfunction under disease conditions.

## Results

### The heterogeneity of the GAIN domain necessitates structure-based residue numbering

A comprehensive analysis of GAIN domains by means of multiple sequence alignments fails due to their low sequence identity and variable number of segments (α-helices and β-strands)[9,10,35,36]. Thus, to enable a comprehensive description of the GAIN domain, we opted for a structure-based approach, for which we generated a set of 14,435 GAIN domain models encompassing orthologs of the 33 mammalian aGPCR and 916 GAIN domains not matched to any orthologs with ColabFold/AlphaFold 2[47,48]. In order to assess the composition of both GAIN subdomains, we used structural alignments with GESAMT[52] for indexing segments. Using this approach, the segment position in space determines its index instead of its sequence-based order, allowing the assignment of equivalent positions for GRN indexing in the context of variable domain composition.

Based on the complete set of GAIN domain models, we asserted that subdomains A and B are composed of two to six helices and 12–14 strands, respectively, which we indexed using the identifiers H1-6 and S1–14 (Fig. 2a). Subdomain B exhibits generally high segment

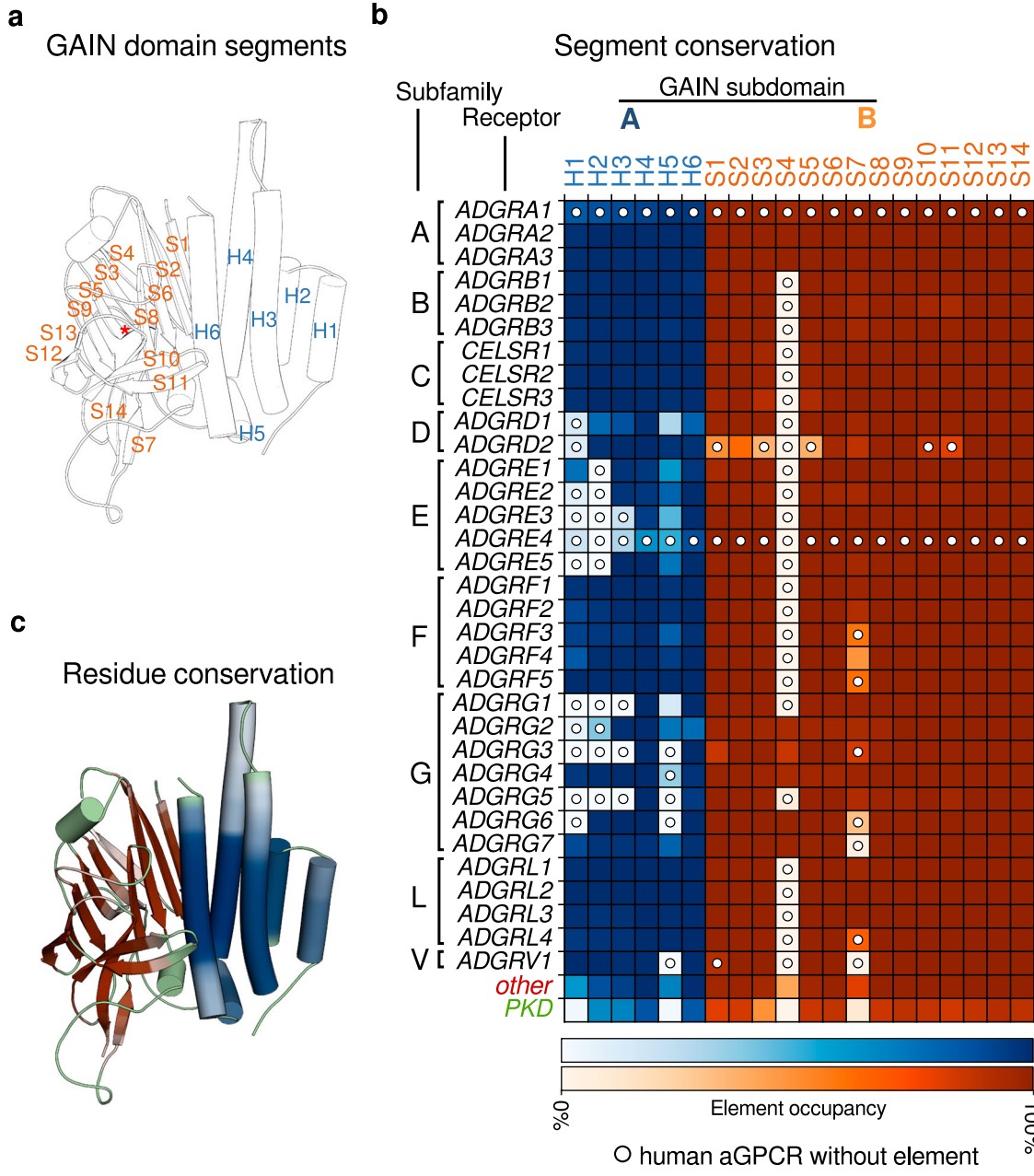

**Fig. 2 | GAIN domain helix and strand architecture of human adhesion G protein-coupled receptors (aGPCRs) and conservation across orthologs.**
**a** Example chimeric GAIN domain structure showing all six helices, 14 strands, and the GPCR proteolytic site (GPS) on its topology, a red asterisk marks the autoproteolytic cleavage site N-terminal of the tethered agonist (S14). **b** each row represents all orthologs in the UniProtKB database that have a GAIN domain for each receptor, where individual elements are highlighted by occupancy (blue: subdomain A helices, orange: subdomain B strands), higher color intensity represents a higher conservation of the element within the group of orthologs. White

circles denote elements that are not present in the corresponding human aGPCR GAIN domain (ADGRA1 without GAIN; ADGRE4 is a pseudogene in humans). Other receptors (red label) are aGPCRs without a receptor ortholog in humans. A set of 2872 polycystic kidney disease-type proteins (PKD, green label) have GAIN domains, which were matched against the set of aGPCR templates, matching well with an additional beta-sandwich subdomain between extended S9 and S10.
**c** residue conservation for residues indexed with the GAIN-GRN for subdomain A (shades of blue) and subdomain B (shades of orange), with 14435 GAIN domains as the underlying number of GAIN domains. Unindexed residues are colored green.

conservation, with only strand 4 specific to subfamilies A and G (Fig. 2b). The composition of subdomain A is more variable. While the A, B, C, F, and L subfamilies all have six helices, the D and G subfamilies show heterogeneity ranging from two to six helices (Fig. 2b). Structures reflecting subdomain A variability are for example the rat ADGRL1 with a six-helix bundle (Fig. 1c)[9], and the human ADGRG1 GAIN domain with only Helix 4 and 6[36], the two most conserved helices in the dataset.

When looking at individual residue positions (to which GRN labels are assigned, Fig. 2c), notably the center regions of helix 3, 4, and 6 are more frequently occupied (occupancy referring to the fraction of

models containing a segment in the dataset) than the extreme positions, highlighting varying helix lengths in the model dataset, with especially the L subfamily exhibiting longer helices. Aside from the residues in the less conserved strands 4 and 7, there is high occupancy in subdomain B. The unindexed GAIN domain loops connecting the structured elements show very different lengths, frequently exceeding 50 residues (Supplementary Fig. 6). Notably, a total of 84 homologs of ADGRA1 in 47 species have a GAIN domain, which is not found in human ADGRA1[9], whereas 78 GAIN domains were identified for ADGRE4, which is a pseudogene in human (Supplementary Table 1).

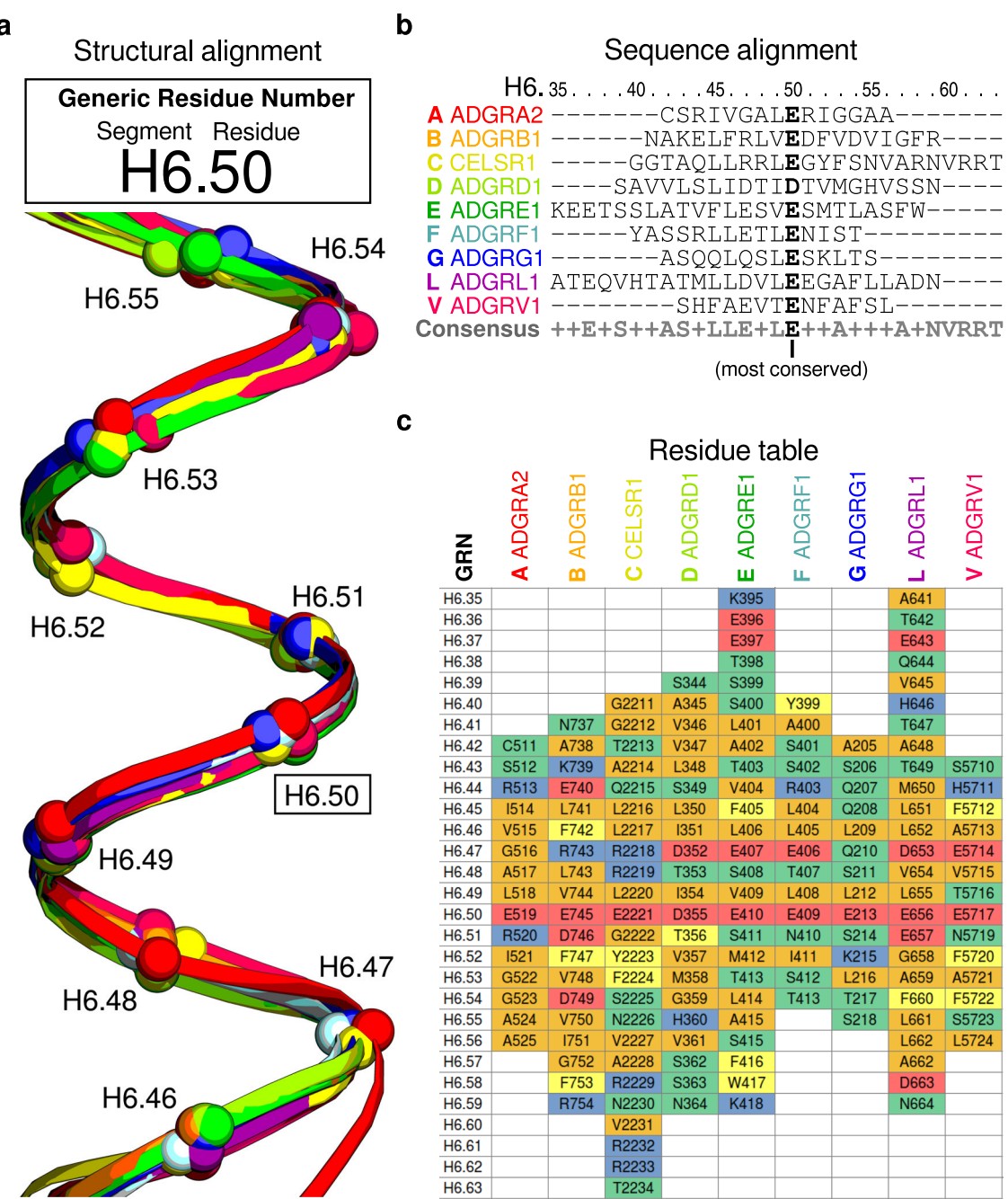

**Fig. 3 | Generic residue numbering denotes corresponding residues across receptors.** Generic residue numbers (GRNs) are equivalent residues in structure and sequence across receptors, enabling comparisons of, e.g., mutations[39,53,85], sequence conservation[39] structural contacts, and ligand interactions[50,85]. **a** A GRN is composed of the segment identifier and the numeric residue index. The structural alignment of the GAIN helix 6 segment of nine human adhesion G protein-coupled receptors indicates the GRN at the respective aligned residue Cα (spheres). **b** sequence alignment of the nine receptors with the receptor subfamily indicated in bold in front of the receptor name. Sequence alignment is based on structural alignment, with the H6.50 GRN denoting the most conserved residue in the segment (bold). All other segment residues are indexed relative to the .50 position. **c** residue table of aligned segments colored by chemical properties.

## Generic residue numbering denotes corresponding GAIN domain amino acids across receptors

Based on the GAIN-GRN indexing of all 14,435 structural models, we created comprehensive alignments of the GAIN domain in structure and in sequence (Supplementary Fig. 1). These provide a novel utility to map data across all adhesion-GPCRs and cross-map sequence-position specific data between homologs. A schematic of the GRN assignment process is outlined in Fig. 3. Each GRN consists of the segment identifier (e. g. Helix 6 = "H6") and the respective index relative to the most conserved residue in the segment, separated by a dot (e.g., "H6.50"). In this example, representative GAIN domains from each aGPCR subfamily are structurally aligned, with the Cα-atoms corresponding to GRN positions (Fig. 3a). Using the backbone alignment positions as the basis for a sequence alignment, the most conserved position is identified here as the acidic E/D and gets assigned the "center" .50 index (Fig. 3b). A residue table of the aligned segments highlights variation in segment lengths and reveals positions with similar physicochemical properties, e.g., the H6.50 as acidic, H6.45, H6.46 and H6.49 as aliphatic, despite low sequence identity (Fig. 3c).

## GPCRdb resources aiding use of GAIN GRNs

As part of the GPCRdb integration, we assigned the 6 helices (H1-6), 14 strands (S1–14), and 21 loops connecting segments (h1h2, h2h3, h3h4, h4h5, h5h6, h6s1, s1s2, s2s3, s3s4, s4s5, s5s6, s6s7, s7s8, s8s9, s9s10, s10s11, s11s12, s12s13, s13gps, gpss14, s14tm1) to all Class B2 (Adhesion) sequences. In addition, the GPS motif has a separate assigned segment from the conventional GPS.-2, GPS.-1, and GPS.+1 notation for the three residues directly preceding or following the

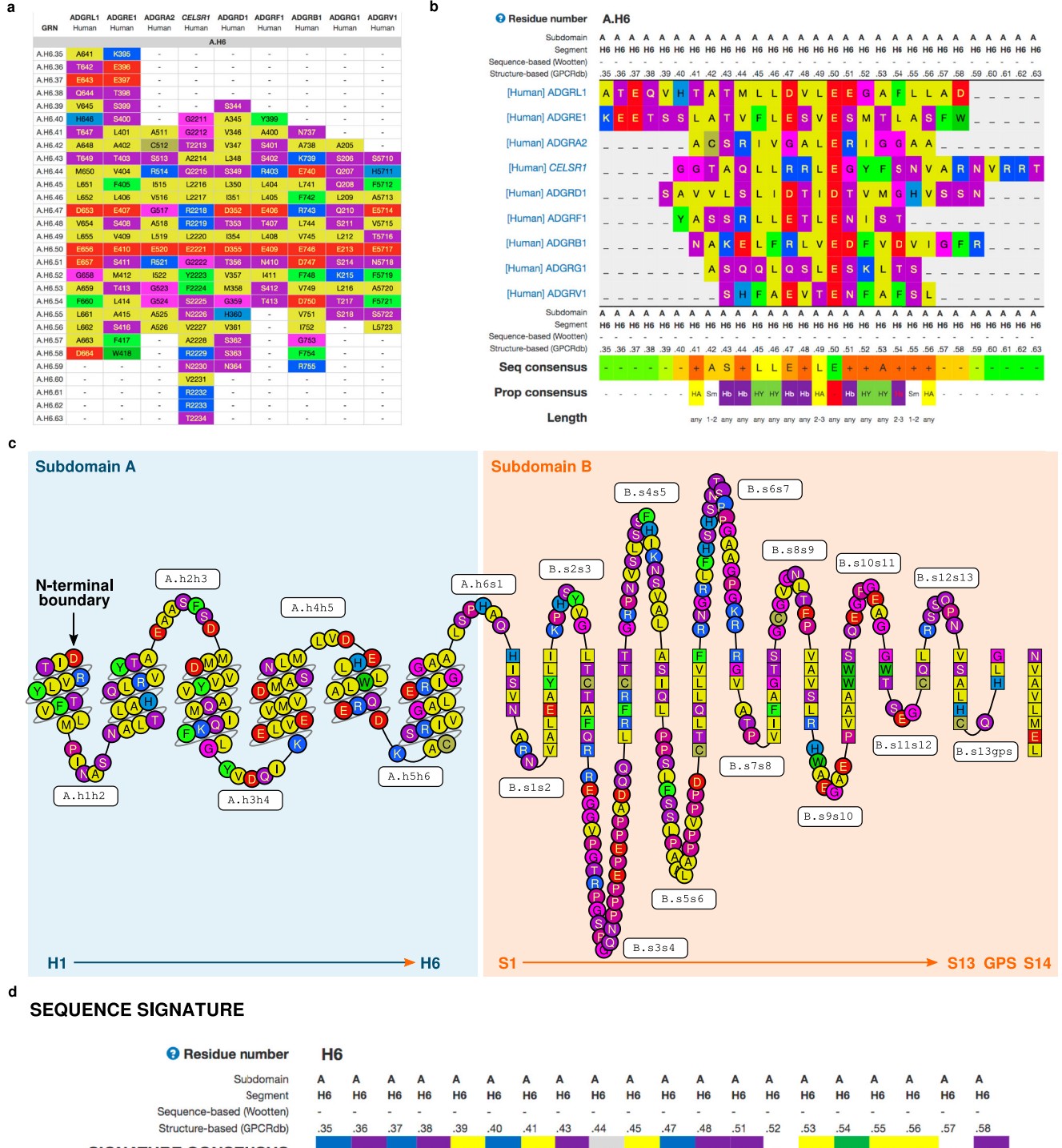

**Fig. 4 | GAIN-GRN data and tools available in the GPCR database, GPCRdb.**
**a** Generic residue numbering tables show GRNs followed by receptor-specific residue numbers and amino acids[53]. **b** Sequence alignments allow swift sequence comparison across all GAIN segments (helices and strands) as well as conservation (% identify and consensus sequence) and physicochemical properties (residue polarity, size, helical propensity and z scales)[50]. **c** the snake plot of the GAIN domain provides a simple 2D representation with the option of custom coloring[53]. **d** The Sequence signature tool identifies structural determinants – uniquely conserved residues – upon contrasting two sets of sequence alignments of receptors that have and lack the given function, respectively[50].

catalytic site. With the introduction of these segments, researchers can apply the GPCRdb toolkit to the whole, or selected parts of the GAIN domain (Fig. 4). We updated the snake plots of the Class B2 (Adhesion) GPCRs to contain the segments of the GAIN domain (Fig. 4c). The snake plots can be found on the Receptor page (https://gpcrdb.org/protein/) with custom coloring options[53]. Along with the already provided data, GAIN domain data is also accessible programmatically via REST API, enabling seamless integration of GAIN-GRN into Python workflows with, i.e., the mdciao package[54].

### Consensus contacts stabilizing the GAIN domain fold
Addressing GAIN domain positions via GRN enables mapping any GRN-label-dependent information across aGPCR homologs. Corroborating the analysis of tertiary structures, we exploit the GRN indexing to consolidate pairwise residue-residue contacts - particular to each structure - into unified consensus contacts. The entirety of GRN-label pairs occurring over the complete dataset with a given frequency yields the GAIN domain contactome, shown in Supplementary Fig. 7 as a flareplot. This plot represents a contact matrix, individually resolving

consensus contacts at the residue level while highlighting contact relationships between the different GAIN segments.

The importance of H6 as a "hub" connecting the subdomains A and B is clearly visible with highly conserved contacts to primarily S6 and also S2, S8, S10, and S14. Furthermore, H4 partially tethers subdomains A and B via highly conserved contacts to S1 and S2. In Supplementary Tables 2 and 3, the most frequent inter-domain and GPS contacts are listed individually, respectively. In addition, we coarse-grained the contactome into the GAIN segments (Supplementary Fig. 8) reproducing the tethering structure of the segments regardless of individual contacts, further highlighting H4 and H6 as the segments mediating contacts between both GAIN subdomains.

### A map of cancer-enriched mutations in the GAIN domain
The GAIN domain, present in 31 of 32 human aGPCRs, is a mutational hotspot affected by various pathologies[7,9,55,56]. To find potential cancer-enriched positions and differentiate them from variance-enriched positions, we adopted the cancer-enrichment score from Wright et al.[57] for all 31 human aGPCR GAIN domains (Fig. 5a, b) indexed by the GAIN-

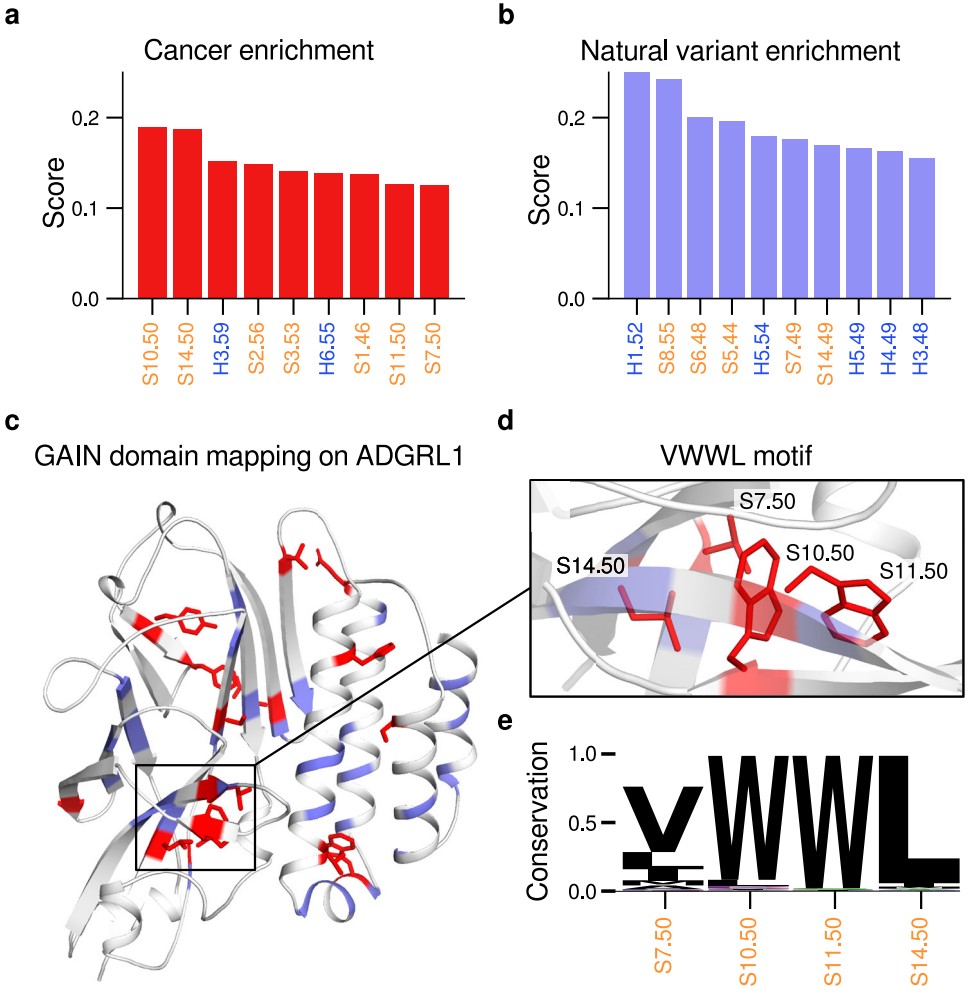

**Fig. 5 | Cancer Genome Atlas mutations mapped onto the GAIN domain via Generic Reisdue Numbers show mutational hotspots.** Enrichment scores were calculated via a relation between the number of naturally occurring variants (retrieved from Genome Aggregation Database (gnomAD), accessed on Jan 16[th], 2023 from https://registry.opendata.aws/broad-gnomad)[86]. and cancer-associated mutations (retrieved from The Cancer Genome Atlas (TCGA) Genomic Data Commons, (GDC), portal.gdc.cancer.gov)[87] at a GRN according to the formula from Wright et al., 2019[57]. **a** The ten most cancer-enriched positions in humans, with the S4 element excluded due to too low a number of mutations and variants. **b** The ten most natural variant enriched positions, showing negative association with cancer,

analogous to the mutation enrichment score[57] **c** enriched positions mapped onto the human ADGRD1 GAIN domain model (UniProt ID: Q6QNK2), with values above 0.1 of maximum intensity colored for cancer enriched (red sticks) and variant enriched (lavender) positions. **d** a cluster of cancer-enriched positions shows the most conserved residues of four strand segments (S7.50, S10.50, S11.50, and S14.50) contacting each other. All positions are part of the ten most cancer-enriched positions. **e** logo plots of residue conservation (fraction of total structures with 1.0 meaning that the position is conserved in all 14,435 GAIN domains in the dataset) for the enriched cluster show strong residue conservation for the VWWL motif composed of V[S7.50], W[S10.50], W[S11.50], L[S14.50].

GRN. While generally, enriched positions of both types are distributed throughout the GAIN domain (Fig. 5c), eight of the ten most cancer-enriched GRN positions are found in subdomain B carrying the TA (Fig. 5a and Supplementary Fig. 2).

We identify a "VWWL" motif consisting of four conserved top-ten cancer-enriched residues S7.50, S10.50, S11.50 and S14.50. This buried motif is located in the direct vicinity of the GPS cleavage site (Fig. 5d–e and Supplementary Fig. 1), with mutations known to affect GAIN domain autoproteolysis and TA function: the conserved leucine at the S14.50 position (Supplementary Fig. 1) is a TA residue deeply buried into the orthosteric binding site of the 7TM domain in active aGPCR-7TM structures[19,30–34,58], and its mutation led to altered receptor activity[11,22,59,60]. Moreover, the mutation of any tryptophane within the "VWWL" motif causes loss of function in rat ADGRL1[9].

With a comprehensive analysis of human aGPCR GAIN domains, we find a total of 46 statistically cancer-enriched positions (Supplementary Fig. 2). By using the GAIN-GRN, homologous cancer-enriched residue positions can now be assigned to any GAIN domain and analyzed for their individual origin and consequence with the tools provided in the repository. This allows the transfer of positional information between GAIN domains in different species, particularly from humans to model organisms, such as *D. melanogaster*, *C. elegans* or *D. rerio*[61–64]. The functional analyses in these model systems can now provide valuable insights into the molecular causes of cancer mutations in humans in future studies.

To further highlight the potential of our approach to evaluate position-specific information, we applied our code repository to conduct a more in-depth sequence and structural bioinformatics analysis of the natural variants neighboring the VWWL motif. The side chains of two natural variance-enriched S7.49 and S14.49 positions, neighboring the cancer-enriched x.50 residues of the VWWL motif point away from the cleavage site. At S7.49 a manifold of amino acid substitutions is found in agreement with its unburied position. By contrast, at the buried S14.49 position, we exclusively find substitutions to hydrophobic residues (Supplementary Fig. 9). Due to solvent exposure the position S7.49 accordingly seems rather insensitive to sequence variations, while at S14.49 variations to non-hydrophobic residues seem not to be tolerated.

## GAIN domains of PKD1/PKD1-like proteins possess an extended topology

The only other protein family known to contain GAIN domains are PKD (polycystic kidney disease)1/PKD1-like proteins (in short here PKD1; also referred to as polycystin-1[PC1])[9]. Mutations in PKD1 are responsible for the majority of autosomal dominant kidney PKD, a devastating disorder that entails the development of cysts in the kidney and other organs leading to their eventual failure[65]. PKD1 GAIN domains display similar molecular properties as aGPCR GAIN domains with autoproteolytic cleavage resulting in a bipartite NTF-CTF protein layout after proteolysis[66]. Enabling the comparison and transfer of experimental and mutational knowledge between aGPCR and PKD1 GAIN folds is the basis for the understanding of similarities and differences between the two, and can offer valuable insights into the cell biological and physiological consequences of GAIN domain functions. However, thus far such transfer has been obstructed by the lack of clear homology assignments of primary and secondary structural positions between aGPCR and PKD1 GAIN domains.

Here, we employed the GAIN-GRN scheme to allocate positional labels in PKD1 GAIN domains and compare them to those of aGPCR GAIN folds. Since no experimental structure of PKD1 GAIN domains is available yet, we prepared 2738 structural models analogously to the aGPCR dataset[66–68]. We applied the GAIN-GRN scheme to the models, which on average resulted in four subdomain A helices and twelve subdomain B strands recognized by the GAIN-GRN method (Fig. 2b), thus structural elements homologous to aGPCR GAIN domains.

Interestingly, we also observed differences to aGPCR GAIN domain layouts as the PKD1 GAIN domains showed an additional β-sandwich fold, which contains an extension of S10 and C-terminally elongated TA. Finally, we also observed up to a total of eight subdomain A helices with additional, unindexed helices (Supplementary Fig. 3).

In sum, the GAIN-GRN scheme provides a robust strategy to allocate structural homologies at the primary and secondary levels also to GAIN folds of PKD1 molecules, which now renders positions in both GAIN domain types comparable to one another.

## Discussion

The GAIN domain is an ancient extracellular protein domain of the large adhesion GPCR family, involved in neural development, hereditary disorders, and cancer[1–4]. Despite recent insights obtained from high-resolution structures of GAIN domains in complex with the 7TM[18], the GAIN domain function in autoproteolysis, mechanosensing, and TA-dependent receptor activation is still poorly understood. To overcome the limitations imposed by the structural heterogeneity of GAIN domains in aGPCR homologs - the variable number of secondary structure segments and overall low sequence identity - we developed the GAIN-GRN as a generic residue numbering scheme for aGPCR GAIN domains. We used spatial alignments of structural models, generating multiple sequence alignments to define the reference residue position as the most conserved residue in each segment[39,50,53]. The GAIN-GRN is based on GAIN domain models predicted by AlphaFold 2/Colabfold to include most GAIN domains in proteins present in the Uniprot database[42,43]. To aid users in employing GAIN GRNs for data analysis and hypothesis generation, we implemented the GAIN-GRN in the GPCRdb serving as an accessible and established resource (Fig. 4). We also show that the GAIN-GRN is a robust tool to assign structure-homologous residues across molecule families as we have retrieved GRN also for PKD1 GAIN domains. Upcoming experimental structures of GAIN domains may have constrictions or other unpredicted structural features that would require future refinement of the GAIN-GRN algorithm, which would be implemented by a new set of structural templates. Notably, the structure-based statistical approach of generating generic residue numbers may also be generalizable by making appropriate adjustments to the source code as outlined in the provided code repository (https://github.com/FloSeu/GAIN-GRN).

Statistical evaluation of the dataset of aGPCR GAIN domains enables us to assess their composition as well as the spatial and positional conservation of information as reflected by the GAIN *contactome* (Supplementary Figs. 7, 8). The evolutionarily conserved two-subdomain architecture of the GAIN domain is present in humans as well as distantly related organisms such as *Trichoplax adhaerens*[5,69–72]. Subdomain B, containing the autoproteolytic cleavage site and the tethered agonist, is structurally less variable consisting of 12–14 β-strands, in agreement with its implied function in NTF-CTF association, force-dependent GAIN domain separation and mechanosensing[26,27,29]. Notably, our analysis underlines the notion that the known "GPS motif" is not an individual protein domain, as initially anticipated, but rather the C-terminal section of subdomain B[8,9]. By contrast, subdomain A, shows high structural heterogeneity with only two critically conserved helices (H4 and H6, Fig. 2), with their core regions forming an interface with and presumably stabilizing subdomain B (Supplementary Figs. 7, 8). Despite structural heterogeneity, our structure-based alignment reveals highly conserved stretches of residues and segments with low overall sequence identity but similar physicochemical properties (Supplementary Fig. 1), thus corroborating the notion that structural conservation outweighs sequence conservation[73].

Creating structure-based alignments of larger protein sets with representatives in humans enables us to structurally map benign and malign mutations for testing in homologous positions of distantly related proteins. For example, the mutations within the "VWWL" motif (Fig. 5) close to the GPS may now be tested in any model system based

on their GRN index. Analogously, we can now assess the location of known pathological mutations: Avila-Zozaya et al. have investigated cancer-related mutations in ADGRL3, with impacts on $G_{13}$-signaling for K561N[H1.51], D798H[S9.47], S810L[s9s10] and E811Q[s9s10], where the latter two residues correspond to the interaction region of the GAIN domain with the seven-transmembrane domain[19,56]. Two mutations responsible for loss of surface expression in GPR56, causing bilateral frontoparietal polymicrogyria (BFPP), are the highly conserved C346S[S10.47] and W349S[S10.50] refs. [7,74,75] More generally, our approach promotes future experiments focusing on the central role of GAIN domains in the physiological functions of aGPCRs and PKD1 molecules and will aid analyses of how structural anomalies contribute to their dysfunction under disease conditions.

## Methods

All computational pipelines were implemented in Python 3.9.16. Packages used for data collection: colabfold 1.2.0, alphafold-colabfold 2.1.6, docker 5.0.3. Packages used for data analysis: stride, ccp4.8.0 with GESAMT 8.0, numpy 1.23.2, matplotlib 3.7.1, Jupyter 1.0.0, Ipy-kernel 6.19.2, Logomaker 0.8, nglview 3.0.4, Scikit-learn 1.3.0, colab-fold 1.2.0, pandas 1.5.3, MAFFT 7.490. Figures were created with The PyMOL Molecular Graphics System, Version 2.2.5, Schrödinger, LLC.

### Generation of the GAIN domain model dataset

Sequences were retrieved from the UniProtKB database with two queries for adhesion GPCR and CELSR, respectively, yielding 22,946 and 2179 sequences, respectively (Supplementary Fig. 5). Sequences were filtered for a minimum length of 50 residues and the presence of a "GPS" domain annotation in their domain records. The C-terminal sequence boundary was read from the "GPS" Domain record, whereas for the N-terminal boundary, sequence lengths exceeding 800 residues were truncated to include the GAIN domain boundary expected at around 320–360 residues upstream of the C-terminal boundary, resulting in 16,537 sequences. The structures of all processed sequences containing potential GAIN domains were predicted with ColabFold[47,48] by using batches of 30 length-sorted sequences with a pre-defined padding to account for sequence length differences per batch Root mean square deviation (RMSD) values of GAIN domains from a respective experimentally determined structure were generally low, with PDB: 4DLQ, rat ADGRL1, RMSD 0.45 Å; PDB: 4DLO, human ADGRB1, RMSD 1.34 Å; PDB: 5KVM, human ADGRG1, RMSD 0.77 Å; PDB: 6V55, danio rerio ADGRG6, RMSD 0.82 Å; PDB: 8IKJ, human ADGRE5, RMSD 0.98 Å; PDB 7QU8, human ADGRG3, RMSD 0.96 Å), and deemed sufficient to keep the modeled structures instead of the experimental ones. A multiple sequence alignment was constructed from an initial 15,957 successfully folded and non-doublet aGPCR/CELSR sequences using MAFFT[76] for localizing the GPS motif.

The secondary structure information of the resulting folded structures was read out with STRIDE[77] The data from the resulting files was used to apply two criteria for a valid GAIN domain: The presence of both the helical subdomain A and the β-sandwich subdomain B as well as the existence of the GPS or a homologously aligned sequence. The filtered dataset consists of 14,435 valid GAIN domains (Supplementary Fig. 5). The human dataset consists of 31 aGPCR GAIN domains.

### GAIN domain detection

The presence of both subdomains was detected by using a numerical transformation of the sequence, assigning a 1 to helical and −1 to β-strand residues. By using linear convolution, a signal was generated, whose sign changes were detected as boundaries between helical and β-strand protein segments. The presence of both subdomains was confirmed by identifying the largest helical segment adjacent to a C-terminal β-strand segment corresponding to the subdomain in each respective structure. The signal decay N-terminal of subdomain A, by

the presence of non-helical residues, was used to determine the GAIN domain boundary for each generated model. The column index of the GPS.-1 residue (corresponding to leucine in the conserved HL | S/T triad) was set as a reference for detecting the presence of a GPS motif or homologous aligned sequence elements. Any structure showing a residue at the corresponding column in the MSA was set as possessing the GPS, therefore satisfying the second criterion.

### Template model selection

Template candidates were extracted by selecting a random 400 structures of each subfamily GAIN domain model to account for variance in the selection and optimize performance. The sub-selection was used to generate a root-mean-square deviation (RMSD) matrix by pairwise alignment applying GESAMT in the CCP4.8 package[52,78], in pairwise alignment mode, on the respective subdomains. Since pair-wise alignments form the basis of the segment identification process and creation of the following MSA, only the pairwise mode of GESAMT was used for code consistency. The matrix was clustered and sorted using agglomerative clustering via the scikit-learn python package[79], and the lowest-RMSD model of the largest cluster was selected as a candidate template. Candidate templates were checked against each receptor sub-selection of the dataset via occupancy (fraction of structures matching the template anchor) and distance (pairwise Cα-Cα distance). Repeating the template selection and curating workflow four times, matched receptors of low quality from the initial set were removed, and additional templates were added and selected from individual receptor selections. After accounting for all conflicts with the alignment functionality in GPCRdb, coverage of all aGPCR protein paralogs, and coverage of spatial orientation variance in individual segments, manual adjustments were made in the form of including strand 4 of subdomain A with 10% occupancy in the model dataset, and an adjustment of the center position in helix 4 to account for the variance in orientation in helix 4, sometimes overlapping with Helix 5. The final set of templates consisted of 15 subdomain A and two sub-domain B templates for the complete indexing. Segment center residues for each element were generated by pairwise aligning all GAIN domains against each candidate template using GESAMT and collecting all pairwise residue matches into a multiple sequence alignment, finding the position of highest occupancy and residue identity. Segment centers were validated and manually curated via 3-D, aligning all candidate templates and verifying the identical position of the anchor in space. The position of the H4 segment center was manually adjusted to avoid ambiguities with the H5 residue center. The unique orientation of the most N-terminal helix of ADGRD1, ADGRE1, and ADGRF4 yielded three individual segment centers. Each receptor GAIN was assigned a template per subdomain to be matched to by default.

### Segment overlap and ambiguity cases

For some cases of low-quality proteins, the SSE of the template and GAIN were overlapping without a pairwise match of the template anchor. In these cases, the match closest to the template anchor was set as the reference position considering the offset (i.e., "S14.47" when the residue is three residues N-terminal of the template S14.50) and enumerated analogously from there.

Anchor ambiguity cases arose when two elements were detected as one by STRIDE with two template center residues matched, however the spatial orientation of two SSE was distinguishable. These cases were handled by a hierarchical segment splitting routine assessing the segment between both matched segment centers in decreasing priority: the presence of a coiled residue, a residue with backbone angles outside of five standard deviations of the element total distribution in the dataset, presence of proline or glycine and a manually defined truncation element for common occurrences.

## Creating the template set

Templates are defined as consensus structural models used for structural alignment of other GAIN domain models for segment identification and indexing. Templates were defined separately for GAIN subdomains A and B. The definition of the template set consisted of three steps: Identifying candidate template structures, finding their center positions, and assessing their coverage and quality for integration into the final template set (Supplementary Fig. 4). Templates have the center residues of each segment already assigned based on structural alignments of all template structures (Fig. 3b).

## Indexing via GAIN-GRN

Each GAIN domain was pairwise aligned to its assigned subdomain A and B template, respectively. GAIN domains not assigned a receptor were structurally aligned to all templates using GESAMT[52], selecting the lowest RMSD template for each subdomain. For each SSE the residue matching the template center was labeled "##.50" with the corresponding element name (H1–H6, S1–14), enumerating all residues in the SSE with numbers decreasing in the N-terminal and increasing in the C-terminal direction. Each ordered residue in the GAIN was assigned a label and exported tabulated. An additional workflow was created in an interactive notebook enabling the assignment of the GAIN-GRN for any protein with an associated model in the alphafoldDB[80] either retrieving the information about the GPS from the Uniprot database or manually defining the C-terminal GAIN boundary.

## Mutation mapping onto GRN positions

Mutations were retrieved from the Cancer Genome Atlas (TCGA, within the Genomic Data Commons https://portal.gdc.cancer.gov/) for each of the 31 human aGPCRs, yielding a total of 6874 individual mutations. A routine was implemented to correct the residue indices of the GAIN domain residues to match the UniProtKB indices. By matching each position, we assigned the GRN to each occurring mutation within the indexed GAIN domain space with a total of 861 mutations, of which 769 mutations were within ordered segments with individual labels. In addition, we implemented a parsing routine to parse the mapped mutations, map the number of mutations and their occurrence onto any GRN-mapped GAIN domain, and filter mutations by the impact metrics SIFT and Polyphen[81,82] to tailor the query routine to the individual purpose. In our example, cancer-enriched positions were extracted by calculating the number of cancer-associated mutations against the number of natural variants extracted from dbSNP (www.nbci.nlm.nih.gov/snp/) analogous to Wright et al., 2019[57].

## Contact frequencies

For each of the 14,435 GAIN domain structures in the dataset, heavy-atom residue-residue contacts were computed using a distance cutoff of 4 Å. All pairs of residues sharing a contact are aggregated into a single contact matrix which is indexed with GRN labels. Some elements of this matrix are shown partially as well as in full in Supplementary Tables 2, 3, and Supplementary Data 1. The computation of contacts, GRN label-handling, and plotting (flareplot and contact-matrix) was done using mdciao[54].

## Reporting summary

Further information on research design is available in the Nature Portfolio Reporting Summary linked to this article.

## Data availability

The generated GAIN domain models generated in this study, alongside all analysis data needed for re-producing all data during the process outlined in the methods have been deposited in the online repository zenodo under accession code 12515545[83]. The PDB entries used in this study are available under the following accession codes: 4DLQ, 4DLO, 5KVM, 6V55, 8IKJ, 7QU8 The UniProtKB entries used in this study can be accessed under the following accession code Q6QNK2 Source data are provided in this paper.

## Code availability

The generated code and interactive notebooks are available under https://github.com/FloSeu/GAIN-GRN and also accessible in the online repository zenodo under accession code 14140353[84].

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

## Acknowledgements

We like to thank Peter Stadler and Franziska Reinhardt of Leipzig University for providing initial GAIN domain multiple sequence alignments and Albert J. Kooistra for his consultations. We thank the Stiftung Charité and the Einstein Foundation for their support (to P.W.H. and G.P.H.). This work was funded by the Deutsche Forschungsgemeinschaft (DFG) through CRC 1423, project number 421152132 (project A06 to T.L., projects C01 and Z04 to P.W.H.), and by grants from the Lundbeck Foundation (R383-2022-306) and Novo Nordisk Foundation (NNF23OC0082561) to D.E.G. We gratefully acknowledge the scientific support and HPC resources provided by the Erlangen National High-Performance Computing Center (NHR@FAU) of the Friedrich-Alexander-Universität Erlangen-Nürnberg (FAU) under the NHR project p101ae NHR funding is provided by federal and Bavarian state authorities. NHR@FAU hardware is partially funded by the German Research Foundation (DFG) - 440719683.

## Author contributions

F.S.: Conceptualization; Methodology; Validation; Investigation; Software; Formal Analysis; Visualization; Writing – original draft; Writing – review and editing; Data Curation. P.W.H.: Conceptualization; Supervision; Funding Acquisition; Validation; Writing – original draft (supporting); Writing – review and editing. G.P.-H. : Conceptualization, Methodology, Validation, Software (supporting), Writing – review and editing; Visualization. G.P.-S.: Software; Resources; Validation; Visualization; Data Curation. R.G.-G. Conceptualization (supporting); Validation (supporting). T.L.: Conceptualization (supporting); Writing – original draft (supporting); Writing – review and editing. D.E.G.: Conceptualization; Supervision; Funding Acquisition; Validation; Writing – original draft (part and supporting); Writing – review and editing.

## Funding

## Competing interests

The authors declare the following competing interests: D.E.G. is a part-time employee and warrant holder at Kvantify. The remaining authors declare no competing interests.
