## [Transparent Peer Review file · Nature Communications]

Generic residue numbering of the GAIN domain of adhesion GPCRs

Corresponding Author: Professor Peter Hildebrand

Version 0:

Reviewer comments:

Reviewer #1

(Remarks to the Author)

The GAIN domains of adhesion GPCRs (aGPCRs) play a critical role in the cleavage and domain-dissociation processes that are together required for aGPCR signaling. It has been difficult to understand the functional roles of GAIN domain residues because the low degree of primary sequence conservation among aGPCR GAIN domain residues makes it difficult to predict their positions and functional properties. This manuscript describes the development of a structure homology-based algorithm designed to permit the structures of aGPCR GAIN domains to be predicted and compared. The authors develop a Generic Residue Numbering system based upon predictions built upon 6 experimentally determined GAIN domain structures and over 14,000 modeled GAIN domain structures. This system permits the helices and strands of the GAIN domain A and B domains (respectively) to be predicted, aligned and compared. The logic motivating the design of algorithmic tool created is well-described and well-justified. The tool has been made publicly available and the manuscript provides a guide to its application. The authors apply the tool to map residues that have been linked to cancer-associated mutations. Through this effort they identify 4 residues in ADGRL1 that appear to participate in the formation of a structural motif even though they each reside on a separate strand. This analysis suggests the potential power of the tool for the generation of experimentally testable hypotheses. The tool has the potential to be of importance to the aGPCR research community and to research who are working to understand the roles of aGPCR in normal biology and in disease.

It is perhaps unfortunate that the present manuscript does not provide experimental or in silico tests of hypotheses that can be generated through application of the tool. The analysis of the cancer-associated mutations that suggested the existence of the VWWL structural motif also included a presentation of non-cancer associated natural variants associated. It appears that two of these natural variants (S7.49 and S14.49) are adjacent to cancer-associated mutations (S7.50 and S14.50). It would be fascinating to understand whether or how these natural and presumably functionally tolerated variants influence the structure and function of the VWWL motif. It is known that loss of either tryptophan in the VWWL causes loss of ADGRL1 function and mutation of S14.50 alters TA-mediated signaling. The paper would be stronger if data could be provided that test whether or how the natural variants affect ADGRL1 function. Ideally such an analysis would include experimental determinations of ADGRL1 activity in a biochemical assay system. Since the current manuscript is entirely computational it could be argued that experimental assessment of receptor function is beyond the scope of the present study. It would nonetheless be informative, however, if the authors could model the effects of the natural variants on the VWWL structure. Do the natural variants alter the predicted VWWF structure? If so, how does this fit with the interpretation that the VWWF structure is functionally significant and that its alteration leads to disease phenotypes. Such an effort could provide insight into the biological significance of the VWWF motif that the authors identify and might help to explain why mutations in adjacent residues are tolerated.

(Remarks on code availability)

Reviewer #2

(Remarks to the Author)

In this study, Seufert and co-workers generated a generic residue numbering (GRN) for the G protein-coupled receptor (GPCR) autoproteolysis inducing (GAIN) domain, which is ubiquitous in adhesion GPCRs (aGPCRs). The GRN was based

on structural alignments of more than 14,000 GAIN domain structures automatically modeled by ColabFold/AlphaFold 2. The numbering scheme will be added to the already existing and widely used GRNs for GPCRs, G proteins, and arrestins in the GPCR database (GPCRdb).

Leaving aside comments on the rather convoluted structure of the text, I would like to focus on the following few points. The inclusion of high-resolution structures of the GAIN domains in the pipeline is mentioned only in the abstract ("based on structural alignment of 6 experimental structures") but never in the main text. How did the authors exploit those structures in their pipeline? Were none of them usable as templates? The choice of representative high-quality templates for pairwise structure-alignments and indexing, a fundamental step, is vague. From over 14000 structural models the authors ultimately identified 15 templates for subdomain A and 2 templates for subdomain B. Template identification passed through RMSD-based clustering of "a random 400 structures of each subfamily GAIN domain models" (why 400?), quality checks (on what basis?), and manual adjustments (in what sense?).

In general, a more rigorous and detailed description of the methodology is needed.

(Remarks on code availability)

Reviewer #3

(Remarks to the Author)

The manuscript details procedure, developed by authors, for uniform numbering of aminoacid residues of the GAIN domain of adhesion GPCRs. This problem is non-trivial because of significant sequence diversity of GAIN domain structures, and can be solved only with a combination of tools and methods. At the same time, the problem needs to be solved to facilitate structural and functional analyses.

The manuscript is written in a clear manner, giving sufficient number of details and is well understood. I particularly liked the substantial introduction into structural biology associated with aGPCR GAIN domains, including examples given in Results and Discussion. I also note rather informative figure legends, which help understanding significantly.

I do not have major comments, and list only minor findings and questions below.

Line 86: "position-specific data transfer": is not clear what it means

Line 100: "non-ortholog": does this mean "paralogs" and if not may be this may be given more clarity

Line 178: "cancer enriched residue positions can be now assigned" I wonder if specific residue types are critical for defining a cancer-enriched position; given considerable sequence diversity, possibly "potential cancer enriched position" would sound better

Line 262: "lengths were truncated" sounds a bit unclear, was it sequences truncated or removed completely from the dataset. Explanation as to why this truncation was necessary would also help

A more general question on the method and choice of GESAMT. This tool performs multiple alignment of structures, which, potentially, could give the sought residue numbering automatically and in a deterministic way. This option, however, was not used; in essence, the authors describe procedure of manually-assisted multiple structure alignment. It would be nice to explain why multiple structure alignment in GESAMT could not be used in this study.

GESAMT was shown to be a highly sensitive and specific tool for structure alignment, however, it disregards not only residue types but also secondary structure patterns. From the context of this study, it seems that structural alignment with focus on SSE match could be rather useful. Such alignments are delivered by several tools, SSM, VAST, DALI to name a few of many. It would be nice to put a few words to justify the use of the particular tool, GESAMT, in this study.

Finally, the problem of assigning consistent and uniform residue numbers is common for many targets in structural biology. If developed software can be applied to other systems, that would be nice to mention in the paper.

(Remarks on code availability)

The code is freely available from github, however with rather minimalistic annotation. A single-line README file is provided, which is not helpful. No build/deployment script, tests or examples are provided. Python files do not contain `__main__` section at the bottom, which is recommended for quick checks and testing. Using this code does not seem to be straightforward.

Reviewer #4

(Remarks to the Author)

This is a review of the paper "Generic residue numbering of the GAIN domain of adhesion GPCRs" by Peter W. Hildebrand and colleagues. The paper introduces a generic residue numbering (GRN) scheme for GPCR auto proteolysis inducing (GAIN) domains of G protein-coupled receptors (GPCRs). The GRN is based on, and can be considered an extension of an already well-established and widely used GRN, originally created for the 7 transmembrane domain of GPCRs, that has since been extended to cover other domains of GPCRs. The authors also claim that the GRN is applicable to GAIN domains from another protein family, polycystic kidney disease 1/PKD1-like proteins.

The GAIN domains covered in the paper are specific to Adhesion-type GPCRs and have very low sequence conservation, but recently published X-ray and Cryo-EM structural information, coupled with advances in protein modeling have allowed structural comparisons of the domains that were previously not possible, showing a conserved fold despite low similarity on the sequence level. The authors build on this recent knowledge and provide a very useful numbering system that enables better comparison of receptors in this family, including mapping of in-vitro mutation data between GAIN domains of different receptors. The authors also make web-based tools to apply the GRN available. Furthermore, the GRN information on GAIN domains is made programmatically available by the authors, and for this, I commend them.

The results presented by the authors and the accompanying web tools will be impactful and enable other researchers in the field.

The manuscript is well written and methods are explained in an understandable manner.

I have no major concerns and support publication of the manuscript. I do however have a few minor concerns and suggestions that could improve the manuscript prior to publication:

1. As the authors describe in the introduction, the existing structure-based GRN of GPCRdb takes into account structural irregularities such as helical bulges and constrictions, assigning both a sequence-based and structure-based numbers to residues where relevant. Were such irregularities observed in any of the GAIN domain structural models? Can the authors address this question and provide illustrated examples of irregularities, if they were found.
2. The results section, position H6.52 is described as "aliphatic" in reference to the table shown in Fig. 3c, but only 5 of the 9 aligned residues are aliphatic. This is therefore a poor example "a position with similar physicochemical properties despite low sequence identity".
3. Fig. 4b shows a snake plot of a 7TM domain, not a GAIN domain as the description says. Can the authors replace the figure with a snake plot of a GAIN domain?
4. The authors refer to a web-based notebook that enables ad-hoc indexing of any GAIN-domain containing protein and provide a URL to a Github repository. The repository contains many files, and it is unclear where the aforementioned notebook is. Could the authors provide a direct URL to the notebook, along with guidance on how to use it, e.g. in the repository's README file?

(Remarks on code availability)

Version 1:

Reviewer comments:

Reviewer #1

(Remarks to the Author)

The authors have responded adequately to the points that I raised in the initial review. In response to a suggestion made in the review they have added new analysis, which strengthens the paper.

(Remarks on code availability)

Reviewer #2

(Remarks to the Author)

The revision brought improvements. I have no additional comments.

(Remarks on code availability)

Reviewer #3

(Remarks to the Author)

My comments from previous round of refereeing have been addressed in full and I do not have further questions and suggestions. A minor typo: CCP4.0 in line 308 should be corrected to specify CCP4 version properly.

(Remarks on code availability)

Reviewer #4

(Remarks to the Author)

In the revised manuscript, the authors have addressed my minor concerns by correcting the text and figures, as well as

restructuring the associated code repository and improving the documentation and instructions for use of the code.

I consider their response to minor concerns 2-4 from the first review acceptable.

In response to minor concern 1 from the first review, the authors provide a good explanation of their methods to detect helical bulges and constrictions and detail their findings. They claim to have added the following sentence to the discussion to account for emerging structural features:

“Upcoming experimental structures of GAIN domains may have constrictions or other unpredicted structural features that would require future refinement of the GAIN-GRN algorithm, which would be implemented by a new set of structural templates.”

However, I cannot find the sentence in the revised manuscript.

I thank the authors for addressing my concerns, and support publication of the manuscript, provided they add the missing sentence to the discussion sections (as the claim to have done the their rebuttal).

(Remarks on code availability)

We like to thank the reviewers for careful examination of our manuscript that helped us to significantly improve the presentation of our method. Please find our reply below:

Reviewer #1 (Remarks to the Author):

The GAIN domains of adhesion GPCRs (aGPCRs) play a critical role in the cleavage and domain-dissociation processes that are together required for aGPCR signaling. It has been difficult to understand the functional roles of GAIN domain residues because the low degree of primary sequence conservation among aGPCR GAIN domain residues makes it difficult to predict their positions and functional properties. This manuscript describes the development of a structure homology-based algorithm designed to permit the structures of aGPCR GAIN domains to be predicted and compared. The authors develop a Generic Residue Numbering system based upon predictions built upon 6 experimentally determined GAIN domain structures and over 14,000 modeled GAIN domain structures. This system permits the helices and strands of the GAIN domain A and B domains (respectively) to be predicted, aligned and compared. The logic motivating the design of algorithmic tool created is well-described and well-justified. The tool has been made publicly available and the manuscript provides a guide to its application.

We thank Reviewer 1 for summarizing the scientific utility and impact of the generic residue numbers as an algorithmic tool mitigating the low sequence conservation and opening for studies prediction and comparison of any aGPCR GAIN domain. We also thank the Reviewer for acknowledging the clear description of and public open access to this tool, as this is surely critical to its future use by the Adhesion GPCR community globally.

The authors apply the tool to map residues that have been linked to cancer-associated mutations. Through this effort they identify 4 residues in ADGRL1 that appear to participate in the formation of a structural motif even though they each reside on a separate strand. This analysis suggests the potential power of the tool for the generation of experimentally testable hypotheses. The tool has the potential to be of importance to the aGPCR research community and to research who are working to understand the roles of aGPCR in normal biology and in disease.

It is perhaps unfortunate that the present manuscript does not provide experimental or in silico tests of hypotheses that can be generated through application of the tool.

At this point we would like to stress that the primary aim of this manuscript is not the generation and subsequent evaluation of new hypotheses, but rather the demonstration of the utility of the GRN framework to promote future experiments. We provide examples of how the generic residue numbers enable mapping of diverse data. Hence, we used existing experimental datasets of genetic and structural information. Whereas the GRNs provide a powerful tool unlocking predictive hypothesis-generation, this deserves - and requires - dedicated original research papers focusing on the idea generation and validation.

For generating experimental hypothesis, we thus provide the code repository with ready-to-use jupyter notebooks allowing for e.g. GRN-label-specific analysis of natural variants and cancer mutations, which provides more depth than the "birds-eye-view" of the current manuscript. To clarify this, we added the following statement to the abstract:

“In summary, our work enables researchers to generate hypothesis and rationalize experiments related to GAIN domain function and pathology.”

This statement emphasizes the potential impact of our work in addition to a reformulated sentence at the very end of our introduction:

“We expect that these will inspire future studies aiming to elucidate molecular mechanism of GAIN domains in the signal transduction and physiological functions of aGPCRs, and will aid analyses on how structural anomalies contribute to aGPCR dysfunction under disease conditions.”

The analysis of the cancer-associated mutations that suggested the existence of the VWWL structural motif also included a presentation of non-cancer associated natural variants associated. It appears that two of these natural variants (S7.49 and S14.49) are adjacent to cancer-associated mutations (S7.50 and S14.50). It would be fascinating to understand whether or how these natural and presumably functionally tolerated variants influence the structure and function of the VWWL motif. It is known that loss of either tryptophan in the VWWL causes loss of ADGRL1 function and mutation of S14.50 alters TA-mediated signaling.

The paper would be stronger if data could be provided that test whether or how the natural variants affect ADGRL1 function. Ideally such an analysis would include experimental determinations of ADGRL1 activity in a biochemical assay system. Since the current manuscript is entirely computational it could be argued that experimental assessment of receptor function is beyond the scope of the present study. It would nonetheless be informative, however, if the authors could model the effects of the natural variants on the VWWL structure. Do the natural variants alter the predicted VWWL structure? If so, how does this fit with the interpretation that the VWWL structure is functionally significant and that its alteration leads to disease phenotypes.

Such an effort could provide insight into the biological significance of the VWWL motif that the authors identify and might help to explain why mutations in adjacent residues are tolerated.

Stimulated by the reviewers' interest to 'model the effects of the natural variants', we made use of the readily available analysis tools from our code-repository to further evaluate the positional variability at the S7.49 and S14.49 positions (Figure *Supplementary Fig. 9d*). This analysis indicates that the natural variants should have low impact on the protein structure and function: In case of S7.49, due to solvent exposure, a high variety of substitutions are tolerated. In case of S14.49, however, only substitutions with similar physicochemical properties (bulk and hydrophobic) are tolerated, presumably because other mutations would affect function (*Supplementary Fig. 9b-d*). Our analysis emphasizes the potential of our scheme to guide future experiments that are needed to investigate the impact of pathological mutations. Of note, for the impact on protein structure, we cannot consider AlphaFold/Colabfold, since the nature of the algorithm will not alter the backbone conformation upon introduction of point mutations.

To directly share this information with the readers, we added a short paragraph at the end of the results section 'A map of cancer-enriched mutations in the GAIN domain':

“To further highlight the potential of our approach to evaluate position specific information, we applied our code-repository to conduct a more in-depth sequence and structural bioinformatics analysis of the natural variants neighboring the VWWL motif. The side-chains of two natural variance enriched S7.49 and S14.49 positions, neighboring the cancer enriched x.50 residues of the VWWL motif, point away from the

cleavage site. At S7.49 a manifold of amino acid substitutions is found in agreement with its unburied position. By contrast, at the buried S14.49 position, we exclusively find substitutions to hydrophobic residues (Supplementary Fig. 9, see below). Due to solvent-exposure the position S7.49 accordingly seems rather insensitive to sequence variations, while at S14.49 variations to non-hydrophobic residues seem not to be tolerated.”

Supplementary Fig. 9: Variant highlights of GRN labels S7.49 and S14.49 show unlikely functional impact compared to the wild-type. **a**, Logoplot of Strand 7 highlighting the high sequence variability at S7.49 with mainly small medium polar or non-aromatic residues A, V, T, S and I; **b**, Logoplot of Strand 14 shows mainly V, I and L at position S14.49 as residues with similar physicochemical character; **c**, GAIN domain model of ADGRA2 with the “VWWL” motif indicated as grey sticks, V^{S14.49} in cyan and R^{S7.49} in magenta. α -C β bond directions are indicated with colored arrows, S7.49 and S14.49 notably point away from the VWWL motif; **d**, consequence of natural variants in S7.49 and S14.49 show mainly similar physicochemical characteristics of residue substitutions, i.e. substitutions of hydrophobic residue S14.49 with other hydrophobic residues.

Reviewer #2 (Remarks to the Author):

In this study, Seufert and co-workers generated a generic residue numbering (GRN) for the G protein-coupled receptor (GPCR) autoproteolysis inducing (GAIN) domain, which is ubiquitous in adhesion GPCRs (aGPCRs). The GRN was based on structural alignments of more than 14,000 GAIN domain structures automatically modeled by ColabFold/AlphaFold 2. The numbering scheme will be added to the already existing and widely used GRNs for GPCRs, G proteins, and arrestins in the GPCR database (GPCRdb).

Leaving aside comments on the rather convoluted structure of the text, I would like to focus on the following few points. The inclusion of high-resolution structures of the GAIN domains in the pipeline is mentioned only in the abstract (“based on structural alignment of 6 experimental structures”) but never in the main text. How did the authors exploit those structures in their pipeline? Were none of them usable as templates?

We thank the reviewer for this remark. Indeed, while we initially intended to include the experimental structures, we found that the respective ColabFold models (that presumably are based on a training data set including most of the experimental structures) are in excellent agreement with the experimental structures (see Table T1). Moreover, statistically, we considered the impact of including six experimental structures out of 14,435 GAIN domains negligible. We have corrected the abstract accordingly and added a sentence to the method section reflecting the high similarity between the modelled and the experimental structures.

Table T1: Root-mean-square deviation of modeled GAIN domain structures with Colabfold to their respective experimental structures.

Receptor	PDB ID	RMSD to Colabfold Model [Å]
rat ADGRL1	4DLQ	0.45
human ADGRB1	4DLO	1.34
human ADGRG1	5KVM	0.77
danio rerio ADGRG6	6V55	0.82
human ADGRE5	8IKJ	0.98
human ADGRG3	7QU8	0.96

The choice of representative high-quality templates for pairwise structure-alignments and indexing, a fundamental step, is vague. From over 14000 structural models the authors ultimately identified 15 templates for subdomain A and 2 templates for subdomain B. Template identification passed through RMSD-based clustering of “a random 400 structures of each subfamily GAIN domain models” (why 400?),

During the template selection, each receptor type (i. e. “ADGRA2”) was assessed separately. The approach of template selection and curation can be traced in full detail by using the jupyter notebooks provided in the github repository under <https://github.com/FloSeu/GAIN-GRN>. We assumed some, but no extreme variations within the sub-selections, which is why we compromised on performance and selection width by selecting 400 structures to be cross-compared with one another. Major outliers would have had surfaced during template curation, which was not the case. To clarify this point raised by the reviewer, we have updated the Methods section as follows:

“Template candidates were extracted by selecting a random 400 structures of each subfamily GAIN domain models to account for variance in the selection and optimize performance. The sub-selection was used to generate a root-mean-square deviation (RMSD) matrix by pairwise alignment applying GESAMT in the CCP4.0 package^{50,76} in pairwise alignment mode on the respective subdomains.”

quality checks (on what basis?),

In the repository, there is a two-piece workflow of identifying and assessing templates. This workflow was run four times, until we get sufficient agreement on the following information:

- Overlap with the alignment functionality in GPCRdb (no residue/position conflicts within the human set of aGPCR GAIN domains)
- Coverage of all paralogs by a minimum number of templates without residue/residue conflict
- Coverage of spatial orientation variance, i.e. Helix 4, where the center residue was moved seven residues upwards to account for an intra-helix kink

For more in-depth explanation and tracing of each individual step in the template selection process, we kindly refer to the README, FAQ and jupyter notebooks in the repository under <https://github.com/FloSeu/GAIN-GRN>

and manual adjustments (in what sense?).

Segment center locations were arbitrarily selected to mirror the highest-quality residue in each segment. Initially, this was only done by conservation (residue identity), but this was insufficient for Helix 4, where we considered different helix orientations for maximum overlap. See the sentence introduced into the methods section below.

In general, a more rigorous and detailed description of the methodology is needed.

We agree with the notion that the methods do not completely reflect all aspects of the computational steps needed for the GAIN-GRN creation process. We have therefore re-designed the code repository alongside a README, Guide and FAQ section and more clearly annotated notebooks to guide the interested user through the complete process of the GAIN-GRN method.

Additionally, we updated the methods section as follows: *“Repeating the template selection and curating workflow four times, badly matched receptors from the initial set were removed and additional templates were added from individual receptor selections. After accounting for all conflicts with the alignment functionality in GPCRdb, coverage of all aGPCR protein paralogs and coverage of spatial orientation variance in individual segments, manual adjustments were made in the form of including Strand 4 of subdomain A with 10% occupancy in the model dataset, and an adjustment of the center position in Helix 4 to account for the variance in orientation in Helix 4, sometimes overlapping with Helix 5. The final set of templates consisted of 15 subdomain A and two subdomain B templates for the complete indexing.”*

Reviewer #3 (Remarks to the Author):

The manuscript details procedure, developed by authors, for uniform numbering of amino acid residues of the GAIN domain of adhesion GPCRs. This problem is non-trivial because of significant sequence diversity of GAIN domain structures, and can be solved only with a combination of tools and methods. At the same time, the problem needs to be solved to facilitate structural and functional analyses.

The manuscript is written in a clear manner, giving sufficient number of details and is well understood. I particularly liked the substantial introduction into structural biology associated with aGPCR GAIN domains, including examples given in Results and Discussion. I also note rather informative figure legends, which help understanding significantly. I do not have major comments, and list only minor findings and questions below.

We appreciate that the reviewer finds our manuscript “written in a clear manner, giving sufficient number of details and is well understood”. We have addressed their minor issues raised below.

Line 86: "position-specific data transfer": is not clear what it means

Has been re-worded to *“data transfer by common GRN-labels”*

Line 100: "non-ortholog": does this mean "paralogs" and if not may be this may be given more clarity

Has been re-worded to *“916 GAIN domains not matched to any orthologs”*

Line 178: "cancer enriched residue positions can be now assigned" I wonder if specific residue types are critical for defining a cancer-enriched position; given considerable sequence diversity, possibly "potential cancer enriched position" would sound better

Indeed, there is no definite proof of a specific GRN-labeled position to be cancer-enriched, since this approach relies on statistics and available data. We have therefore included the word “statistically” in the phrase. It should remain up to future wet-lab approaches to validate cancer-specific mutations in specific proteins.

Line 262: "lengths were truncated" sounds a bit unclear, was it sequences truncated or removed completely from the dataset. Explanation as to why this truncation was necessary would also help

We have re-worded for more clarity:

“The C-terminal sequence boundary was read from the “GPS” Domain record, whereas for the N-terminal boundary, lengths exceeding 800 residues were truncated to include the GAIN domain boundary expected at around 320-360 residues upstream of the C-terminal boundary, resulting in 16,537 sequences.”

The truncation was necessary since folding too many residues would overflow and crash the folding method. This is the reason why full-length models of huge proteins are not yet available in the AlphaFoldDB, notably ADGRC1-3 and ADGRV1.

A more general question on the method and choice of GESAMT. This tool performs multiple alignment of structures, which, potentially, could give the sought residue numbering automatically and in a deterministic way. This option, however, was not used; in essence, the authors describe procedure of manually-assisted multiple structure alignment. It would be nice to explain why multiple structure alignment in GESAMT could not be used in this study.

The multiple-sequence mode of GESAMT was tested, however deemed impractical for the size of the dataset (14,435 structures). While benchmarking, we found that GESAMT calculation increases $O(n^2)$ when including multiple structures and runs with >400 structures had to be terminated because calculation took far too long to finish during benchmarking. To avoid clunky batch-mode inclusion of multiple GESAMT-MSAs, we used pairwise alignment files with reasonable performance (single-core pairwise alignment of 14,435 GAIN subdomain in <1 minute).

GESAMT was shown to be a highly sensitive and specific tool for structure alignment, however, it disregards not only residue types but also secondary structure patterns. From the context of this study, it seems that structural alignment with focus on SSE match could be rather useful. Such alignments are delivered by several tools, SSM, VAST, DALI to name a few of many. It would be nice to put a few words to justify the use of the particular tool, GESAMT, in this study.

In the workflow, the secondary structure data extracted with STRIDE is used for filtering and validation of the GAIN domain models. With the secondary structure data already present, a barebones approach of pairwise structure alignment via GESAMT was deemed sufficient for the matching the respective structure pairs. We did not want to introduce further bias by matching SSEs, which is why we resorted to a more “barebones” approach of residue-pair matching.

We updated the methods section for the raised previous two points as follows:

“Template candidates were extracted by selecting a random 400 structures of each subfamily GAIN domain models and generating a root-mean-square deviation (RMSD) matrix by pairwise alignment using GESAMT in the CCP4.0 package^{50,76} in pairwise alignment mode on the respective subdomain. Since pairwise alignments form the basis of the segment identification process and creation of following MSA, only the pairwise mode of GESAMT was used for code consistency.”

Finally, the problem of assigning consistent and uniform residue numbers is common for many targets in structural biology. If developed software can be applied to other systems, that would be nice to mention in the paper.

The developed software can be applied to other systems in a modified form. We have used the annotation of the GPS motif to find the C-terminal boundary and signal convolution to identify the N-terminal boundary of the GAIN domain. With a set of clearly defined domains, the workflow described in the GAIN-GRN repository GUIDE and FAQ (<https://github.com/FloSeu/GAIN-GRN>) may be applied analogously, however our subdomain-specific method of GAIN splitting for templates will require some customization to generalize. We have specified a more nuanced set of adjustments in the FAQ file and added a sentence at the end of the first paragraph of the conclusion:

“The structure-based statistical approach of generating generic residue numbers may also be generalizable by making appropriate adjustments to the source code as outlined in the provided code repository.”

Reviewer #3 (Remarks on code availability):

The code is freely available from github, however with rather minimalistic annotation. A single-line README file is provided, which is not helpful. No build/deployment script, tests or examples are provided. Python files do not contain `__main__` section at the bottom, which is recommended for quick checks and testing. Using this code does not seem to be straightforward.

We thank the reviewer for this remark. We have extensively re-factored and re-structured the GAIN-GRN repository (<https://github.com/FloSeu/GAIN-GRN>) to provide a more accessible and transparent experience for the interested user. It is now installable locally via **pip** and employs a README alongside numbered and clearer jupyter notebooks to guide the user through the process of the full generation of the GAIN-GRN dataset, templates and assignment. Scripts for assigning the GAIN-GRN dynamically are also included.

Reviewer #4 (Remarks to the Author):

This is a review of the paper “Generic residue numbering of the GAIN domain of adhesion GPCRs” by Peter W. Hildebrand and colleagues. The paper introduces a generic residue numbering (GRN) scheme for GPCR auto proteolysis inducing (GAIN) domains of G protein-coupled receptors (GPCRs). The GRN is based on, and can be considered an extension of an already well-established and widely used GRN, originally created for the 7 transmembrane domain of GPCRs, that has since been extended to cover other domains of GPCRs. The authors also claim that the GRN is applicable to GAIN domains from another protein family, polycystic kidney disease 1/PKD1-like proteins.

The GAIN domains covered in the paper are specific to Adhesion-type GPCRs and have very low sequence conservation, but recently published X-ray and Cryo-EM structural information, coupled with

advances in protein modeling have allowed structural comparisons of the domains that were previously not possible, showing a conserved fold despite low similarity on the sequence level. The authors build on this recent knowledge and provide a very useful numbering system that enables better comparison of receptors in this family, including mapping of in-vitro mutation data between GAIN domains of different receptors. The authors also make web-based tools to apply the GRN available. Furthermore, the GRN information on GAIN domains is made programmatically available by the authors, and for this, I commend them.

The results presented by the authors and the accompanying web tools will be impactful and enable other researchers in the field.

The manuscript is well written and methods are explained in an understandable manner.

We appreciate that the reviewer finds our numbering system “useful” and our web tools “impactful” and to “enable other researchers” and our manuscript “well written”. We have addressed the minor issues raised by this reviewer below.

I have no major concerns and support publication of the manuscript. I do however have a few minor concerns and suggestions that could improve the manuscript prior to publication:

1. As the authors describe in the introduction, the existing structure-based GRN of GPCRdb takes into account structural irregularities such as helical bulges and constrictions, assigning both a sequence-based and structure-based numbers to residues where relevant. Were such irregularities observed in any of the GAIN domain structural models? Can the authors address this question and provide illustrated examples of irregularities, if they were found.

The question of the presence of bulges is indeed relevant, as it posed significant challenges in assigning generic residue numbers to 7-transmembrane domains of GPCRs. Here, we first examined the structures matched to the subdomain A templates for the content of helical, 3_{10} -helical and outlier helical residues detected by STRIDE (Figure R1). Taking three templates (L4, top; D, center; G7, bottom) as an example, we mostly see variance in H1, H3 and H5, while the secondary structures of other segments remain rather homogenous.

Figure R1: Distribution of secondary structure for matches against Subdomain A templates G7, L4 and D with blue denoting regular α -helices, light blue denoting statistical outliers of α -helices and orange denoting 3_{10} -helix residues. Bulges would be indicated by π -helices, which were not detected.

For the individual detection of bulge candidates, we used the database of pairwise GESAMT alignments of targets to templates. A bulge would be detected by either recognizing a π -helical (constricted) turn in the secondary structure detection or finding gaps in the pairwise residue matches. First, we searched for residues without a template pair match for each helical residue. Second, we deleted unmatched stretches where it was simply the start or end of the helix that was unmatched, leaving us with a total of 2,210 unmatched residue stretches. These residue stretches were grouped and stacked by using the GRN-labels to see how high the frequency of a gap per GRN label and per receptor was (i.e. H6.55 in ADGRA1). Frequencies of below 3% were excluded. Third, the GESAMT files were examined around the gapped region. Where the template was helical both up- and downstream of the gapped region, these pairs were collected, with a total of 327 structure-template pairs to be left for manual examination. Notably, these 327 groupings were composed of only six receptors and their templates (E4:E1, F3:F5, G1:G1, G2:G7, G3:G3 and G4:G7), of which examples are shown in Figure R2 with the putative gap residue highlighted. We can see that while the helix orientation deviates between template and target, these deviations are kinks instead of bulges, i. e. the two helices do not align both up- and downstream of the residue in question. While the quality of those helix-helix matches is rather low, we only match the segment center (xx.50 residue) and enumerate from there to account for larger deviations outside of the most conserved segment region.

We acknowledge that the current GAIN domain models are only corroborated by the presence of six GAIN domain structures in the PDB and represent the best models to-date for generating the GAIN-GRN. To account for emerging structural features, we have added the following sentence to the discussion:

“Upcoming experimental structures of GAIN domains may have constrictions or other unpredicted structural features that would require future refinement of the GAIN-GRN algorithm, which would be implemented by a new set of structural templates.”

Figure R2: Examples for the six receptor-template pairs with bulge candidates show deviations in backbone, yet no bulges. The residue that has no pairwise match partner in the GESAMT pairwise alignment is highlighted in magenta. Template PDB in white, receptor PDB in grey.

2. The results section, position H6.52 is described as “aliphatic” in reference to the table shown in Fig. 3c, but only 5 of the 9 aligned residues are aliphatic. This is therefore a poor example “a position with similar physicochemical properties despite low sequence identity”.

We have substituted H6.52 by H6.49, where the aliphatic character is more evident, evident from the complete alignment panel of H6 in Supplementary Figure 1.

3. Fig. 4b shows a snake plot of a 7TM domain, not a GAIN domain as the description says. Can the authors replace the figure with a snake plot of a GAIN domain?

We have updated Figure 4 to include clearer overview of the GPCRdb functionalities implemented alongside a snakeplot of a complete GAIN domain. Panel 4a has been removed since the dynamic GAIN-GRN assignment notebook is now part of the package and its workflow is therefore not an “online tool”.

4. The authors refer to a web-based notebook that enables ad-hoc indexing of any GAIN-domain containing protein and provide a URL to a Github repository. The repository contains many files, and it is unclear where the aforementioned notebook is. Could the authors provide a direct URL to the notebook, along with guidance on how to use it, e.g. in the repository’s README file?

The structure, file names and annotations in the repository have been updated to provide more clarity into the functionality and purpose of each containing file and making the structure match a python package. Furthermore, a README.rst with installation instructions, a GUIDE.rst for usage and FAQ.rst with additional information have been added to further introduce functionalities and related questions of the code alongside installation instructions. The notebook, due to restrictions in hosting, is available as part of the package and can be installed locally.

REVIEWERS' COMMENTS

Reviewer #1 (Remarks to the Author):

The authors have responded adequately to the points that I raised in the initial review. In response to a suggestion made in the review they have added new analysis, which strengthens the paper.

Reviewer #2 (Remarks to the Author):

The revision brought improvements. I have no additional comments.

Reviewer #3 (Remarks to the Author):

My comments from previous round of refereeing have been addressed in full and I do not have further questions and suggestions. A minor typo: CCP4.0 in line 308 should be corrected to specify CCP4 version properly.

The version CCP4.8 has been corrected.

Reviewer #4 (Remarks to the Author):

In the revised manuscript, the authors have addressed my minor concerns by correcting the text and figures, as well as restructuring the associated code repository and improving the documentation and instructions for use of the code.

I consider their response to minor concerns 2-4 from the first review acceptable.

In response to minor concern 1 from the first review, the authors provide a good explanation of their methods to detect helical bulges and constrictions and detail their findings. They claim to have added the following sentence to the discussion to account for emerging structural features:

“Upcoming experimental structures of GAIN domains may have constrictions or other unpredicted structural features that would require future refinement of the GAIN-GRN algorithm, which would be implemented by a new set of structural templates.”

The sentence in question has been added to the discussion. We apologize for the oversight.

However, I cannot find the sentence in the revised manuscript.

I thank the authors for addressing my concerns, and support publication of the manuscript, provided they add the missing sentence to the discussion sections (as the claim to have done the their rebuttal).